# The Effect of Light Cycles on the Predation Characteristics of *Phytoseiulus persimilis* (Acari: Phytoseiidae) Feeding on *Tetranychus urticae* (Acari: Tetranychidae)

**DOI:** 10.3390/plants14050687

**Published:** 2025-02-23

**Authors:** Hajar Pakyari, Rostislav Zemek

**Affiliations:** 1Department of Plant Protection, Takestan Branch, Islamic Azad University, Takestan P.O. Box 34819-49479, Iran; 2Institute of Entomology, Biology Centre CAS, 370 05 Ceske Budejovice, Czech Republic; rosta@entu.cas.cz

**Keywords:** two-spotted spider mite, predatory mite, predation rate, photoperiod duration, abiotic conditions, biological control

## Abstract

Environmental factors, such as the duration of daylight, can significantly influence the predation ability of arthropod predators. This study aimed to examine the influence of photoperiods of 8:16 h, 12:12 h, and 16:8 h (L:D) on the predation rate of *Phytoseiulus persimilis* preying on *Tetranychus urticae* eggs under constant temperature. The daily predation rate (*D_j_*) and the total number of prey eggs consumed (*P_j_*) per predator increased with longer photophases, reaching their peak in the 16L:8D photoperiod. The highest net predation rate (*C*_0_) was observed under 16L:8D conditions (173.22 prey eggs/predator), while it was 170.28 and 89.77 prey eggs/predator under the 12L:12D and 8L:16D photoperiods, respectively. The finite predation rate (ω) also increased with longer photophases. The transformation rate (*Q_p_*) was highest under the 16L:8D photoperiod. Significant differences were noted in the stable predation rate (ψ), with the highest value being 5.84 prey eggs/predator under 16L:8D conditions. The number of *T. urticae* eggs predated by *P. persimilis* was higher under longer photoperiods, and the 16L:8D photoperiod can thus be recommended as optimal for the biocontrol of spider mites in controlled environments. We suggest that future research explores other effects of the light cycle on plant–herbivore–predator interactions to optimize the lighting conditions for effective spider mite control.

## 1. Introduction

*Phytoseiulus persimilis* Athias-Henriot (Acari: Phytoseiidae) is an important biocontrol agent as a specialist predator of spider mites, particularly *Tetranychus urticae* Koch (Acari: Tetranychidae) [1,2]. This natural enemy is employed globally in various crops to manage mite pest populations [1,3]. Although phytoseiid mites are not as voracious as insect predators, their short life cycle, high reproduction rate, and ease of mass-rearing make them a crucial species for the biocontrol of mite and insect pests in integrated pest management (IPM) programs [2,4,5]. *Phytoseiulus persimilis,* which is present throughout the year, easy to rear and maintain, and suitable for mass production, has been widely used for biological pest control since 1968 and currently is one of the most important arthropod biocontrol agents [6]. The successful application of *P. persimilis* as a biocontrol agent against *T. urticae* has been documented for greenhouse [7,8,9] and field crops [10,11,12,13].

Five developmental stages have been identified in *P. persimilis*: egg, larva, protonymph, deutonymph, and adult. The longevity of mites is influenced by prey stages and abiotic conditions such as temperature and photoperiod [14,15,16]. Several studies have examined the ecology and behavior of *P. persimilis*, including its life table parameters, functional responses, predation rate, and the effects of temperature on its development and reproduction, e.g., [2,15,17,18]. The predation capacity of *P. persimilis* was studied under constant conditions by Moghadasi et al., who reported that this mite species can kill about 700 *T. urticae* eggs during its life cycle [2].

Various factors can influence the activity of predators, including environmental conditions [19], the moon cycle [20,21,22], host plants [23], diet quality [24], the availability of supplementary food [25], prey age [26], and predator body size [27]. Abiotic conditions, such as the length of daylight, are known to affect the predation rate of various biocontrol agents including phytoseiid mites [19,28,29,30,31,32,33]. Assessing the biological performance of these natural enemies under various conditions is essential for evaluating their biological and behavioral attributes [34]. This is particularly important for developing effective IPM programs for crops grown in protected environments, e.g., greenhouses, where light durations can be controlled. While typical greenhouse settings range from 12 to 16 h of light per day, depending on the species being cultivated, some crops may benefit from extended light periods (up to 18–24 h) during specific growth stages, such as vegetative growth, to enhance photosynthesis and biomass accumulation. Hence, understanding the relationship between light duration and plant–herbivore–predator interactions can help in the optimization of biological control strategies.

The photoperiod is a predictable variable throughout the year. It serves as a fundamental signal of seasonal changes, triggering synchronization in the development and reproduction of arthropod species during optimal periods [35]. Various developmental stages of mites and insects are known to respond to changes in photoperiod. Additionally, the photoperiod is a crucial regulatory factor for diapause [36]. Although phytoseiid mites are blind, they can respond to light, which suggests that they have some non-visual mechanisms by which to detect light and darkness. Different photoperiods can affect their life history traits such as their development, reproduction, and survival [16]. This implies that the photophase–scotophase ratio plays an important role in regulating the biological cycles of phytoseiid mites. For example, shorter days may signal changes in season, leading to different behaviors or reproductive strategies. Numerous studies have shown the impact of the photoperiod on various biological aspects of predatory insects and mites, such as fecundity and demographic parameters [16,35,37]. The photoperiod also influences the predation rates of mites. Some species of phytoseiid mites, e.g., *Galendromus occidentalis* (Nesbitt), prefer predation in light [31]. Similarly, *Amblyseius womersleyi* (Schicha) consumed more *T. urticae* eggs in the light than in the dark [38]. This behavior might be related to the specific prey they target, their habitat, or their evolutionary adaptations. However, to our knowledge, information on the effects of photoperiods on *P. persimilis* predation behavior is currently unavailable.

The present study aimed to examine the influence of photoperiods on the predation parameters of *P. persimilis*. While life table parameters providing comprehensive information on the biological traits of this species have been reported recently [16], understanding their connection with predation rates is crucial for estimating the potential of this biocontrol agent used in IPM programs [39,40]. We tested the hypothesis that the length of the photophase significantly affects *P. persimilis* predation parameters. The obtained results are intended to determine the optimal photoperiod for maximizing the predation rate of this predatory mite for release as a biological control in vegetable greenhouses.

## 2. Results

The daily predation rate (*D_j_*) of *P. persimilis* increased with the photoperiod (Table 1), with the maximum values of the *D_j_* observed under 16L:8D conditions for each developmental stage. The total number of prey consumed (*P_j_*) during the preadult stage was 15.65, 18.83, and 22.00 per predator under the 8L:16D, 12L:12D, and 16L:8D photoperiods, respectively (Table 2). The maximum *P_j_* value in adult females was found under the 16L:8D condition (282.22 prey/predator), followed by the 12L:12D (261.29 prey/predator) and 8L:16D photoperiods (145.16 prey/predator).

Table 3 shows the predation parameters of *P. persimilis* under different photoperiods. The maximum value of the net predation rate (*C*_0_) was observed under the 16L:8D (173.22 prey/predator) and 12L:12D (170.28 prey/predator) regimes. The finite predation rate (ω) increased with the length of the photophase. The highest transformation rate (*Q_p_*) was found under 16L:8D conditions, where *P. persimilis* needed to consume 14.37 prey to produce one egg. As for the stable predation rate (ψ), significant differences were also observed, with the highest value (5.84 prey/predator) occurring under long-day conditions.

Figure 1 illustrates the effect of the photoperiod on the age-specific predation rate (*k_x_*) and age-specific net predation rate (*q_x_*) of *P. persimilis* fed on *T. urticae* eggs. Due to the influences of *l_x_*, the values calculated for the *q_x_* differed from those estimated for the *k_x_* under the same photoperiod (Figure 1). The *k_x_* values increased with the age of *P. persimilis*, peaking at ages 17, 19, and 16 days, with values of 17.15, 20.17, and 24.82 prey, respectively, under the 8L:16D, 12L:12D, and 16L:8D photoperiods (Figure 1). Figure 2 shows the age–stage-specific predation rate (*C_xj_*) for different life stages of *P. persimilis* fed on the eggs of *T. urticae* under different photoperiods. This parameter represents the average number of prey consumed by *P. persimilis* at age *x* and stage *j*. Because the eggs exhibited no predation activity and the larvae had minimal feeding activity, the predation rates during immature development were primarily attributed to the two nymphal stages. Adult male predators exhibited relatively lower predation rates compared to females in all treatments. The maximum number of prey consumed by female *P. persimilis* was 18.90, 22.05, and 27.06 at ages 16, 15, and 16 days under the 8L:16D, 12L:12D, and 16L:8D photoperiods, respectively (Figure 2).

## 3. Discussion

In general, *P. persimilis* consumed more *T. urticae* eggs under longer photoperiods. Our previous study demonstrated that extended photoperiods significantly improved the development and fecundity of *P. persimilis*, as well as its population growth rate and finite rate of increase [16]. Enhanced growth under longer photoperiods likely contributed to its predation ability, resulting in higher predation rates. The photoperiod has been widely reported to influence the predation capacity of predatory insects and mites. For instance, Wang et al. [19] demonstrated that prey consumption by fourth-instar larvae and adults of *Cheilomenes sexmaculata* (F.) (Coleoptera: Coccinellidae) increased significantly with longer photoperiods. Tan et al. [28] found that the prey consumption rate of *Hippodamia variegata* (Goeze) (Coleoptera: Coccinellidae) also increased with extended day lengths. Additionally, Kumar and Omkar [29] showed that the highest prey consumption rates for two other coccinellid predators, *Coccinella septempunctata* L. and *Coccinella transversalis* Fab., occurred under a 16L:8D photoperiod. These findings underscore the significant impact of the photoperiod on the feeding behavior and efficiency of predatory insects. Yu et al. [30] demonstrated that *Harmonia axyridis* Palles (Coleoptera: Coccinellidae) predates higher numbers of *Myzus persicae* Sulzer (Hemiptera: Aphididae) under long photoperiods compared to short photoperiods. Currently, limited information is available regarding the influence of the photoperiod on the predation parameters of predatory mites. For instance, Kazak et al. [31] reported that the highest total average number of *T. urticae* eggs consumed by *G. occidentalis* was observed in the 14L:10D photoperiod, but the authors concluded that changing the length of the light–dark periods had no consistent effect on total egg consumption. On the other hand, *Neoseiulus fallacis* (Garman) females were significantly more efficient predators under a 10L:14D photoperiod than mites held at photoperiods of 0L:24D, 14L:10D, and 24L:0D [32], while the predation capacity of *Amblyseius swirskii* Athias-Henriot was higher under complete darkness [33]. However, to our knowledge, no comparable data are currently available for *P. persimilis*.

A light environment with long periods of bright light may aid predators in finding and capturing their prey, indicating that increased light supplementation does not impair their predation ability. Perez-Sayas et al. [41] showed that predation behavior in *P. persimilis* varied daily and seasonally, with peak activity occurring during the daytime. Similarly, the results of the present study showed that both the mean daily predation rates (*D_j_*) and total prey consumed (*P_j_*) by nymph and adult *P. persimilis* feeding on *T. urticae* eggs increased with longer photoperiods (Table 1 and Table 2). Female *P. persimilis* exhibited higher predation rates compared to males, likely due to their greater energy demands during reproduction. Egg production requires substantial nutrition, which may explain the higher predation rates in females. In contrast, males likely require fewer resources to meet their metabolic needs, resulting in lower predation rates. Additionally, males may allocate more energy and time to reproductive activities, such as mate searching, rather than predation [42]. Elucidating the mechanism behind the effect of long days on the feeding behavior of *P. persimilis* will, however, require further experiments. Unlike the ladybirds mentioned above, in which visual cues are likely responsible for their increased predation in response to longer periods of light, the causes of increased predation in phytoseiid mites under longer light periods are less clear. The plant-mediated effect of light may be one of the mechanisms, as indicated by Maeda et al. [38], who found that *A. womersleyi*’s behavior coincided with the production pattern of *T. urticae*-induced plant volatiles, which are mainly produced in the light. The finite predation rate (ω) incorporates parameters such as the finite rate of increase (λ), age–stage-specific structure (*a_xj_*), and age–stage predation rate (*k_x_*), making it a reliable metric for evaluating and comparing the potential of predators as biocontrol agents [43]. This parameter effectively assesses the predation capabilities of natural enemies [39]. This parameter revealed an increase with longer photoperiods in the present study, consistent with findings from previous studies [19,28,29,30]. Such studies establish standard parameters for effectively determining growth and development, stage differentiation, reproduction, and predatory rates in relation to age, stage, and sex [44,45]. However, population parameters alone cannot fully describe the biological performance of predators [40]. Predation parameters are also valuable indicators of the predation capacity of natural enemies under specific conditions. In the current study, the transformation rate (*Q_p_*), representing the conversion of the prey population into *P*. *persimilis* offspring, ranged from 7.99 under a 12L:12D photoperiod to 14.37 under a 16L:8D photoperiod. This parameter provides a demographic measure of the relationship between the reproduction rate (*R*_0_) and predation rate (*C*_0_) of the predator, indicating the efficiency of the prey population’s conversion into predator offspring [23]. The higher transformation rate observed under the 16L:8D photoperiod suggests that individuals achieve maximum fitness under these conditions. This finding is thus important for optimizing conditions for the commercial mass rearing of *P. persimilis*.

Analyzing how the predation rate of natural enemies responds to the photoperiod can significantly enhance the effectiveness of biological control strategies when releasing natural enemies for pest management. This study provides a comprehensive assessment of the predation rate of *P. persimilis* in response to different photoperiods. The results obtained indicate that the number of *T*. *urticae* eggs consumed by *P. persimilis* is higher under longer photoperiods compared to shorter ones. A long-day (16L:8D) photoperiod was identified as optimal for the release of this biocontrol agent against two-spotted spider mites in greenhouse conditions, where managing the duration of light is possible. This photoperiod not only enhances the predation rate of *P. persimilis* but is also disadvantageous for *T. urticae* development [46] and can thus be recommended as an effective IPM strategy for the management of *T. urticae*. On the other hand, short-day conditions inducing diapause in spider mites [47,48,49,50] represent another strategy. A short-day photoperiod (12L:12D) was also reported to lead to the highest population growth of *P. persimilis* [16], which could thus be recommended for the mass rearing of this predator in particular. Besides the photoperiod, light wavelength is known to affect both *T. urticae* [49] and *P. persimilis* [51], opening up other possibilities for spider mite control. A recent study by Savi et al. [52] demonstrated how timed regimes of red, blue, or far-red light-emitting diodes (LEDs) during the day or at night may affect plant traits and the performance of *T. urticae* and *P. persimilis* and concluded that timed LED regimes could be crucial in designing IPM strategies that promote both plant growth and effective biological control in controlled environments.

Future research is certainly needed to explore other effects of the light cycle on plant–prey–predator interactions to optimize lighting conditions for effective spider mite control. This research should focus on, for example, investigating the indirect, plant-mediated effects of photoperiods on *T*. *urticae*, such as those resulting from plant feeding or volatile cues. Additionally, it would be valuable to examine how the performance (i.e., life table parameters and population dynamics) of natural enemies is influenced by the indirect effects of the photoperiod, particularly those mediated through herbivores.

## 4. Materials and Methods

### 4.1. Rearing of Prey and Predators

The mite cultures were established from *P. persimilis* and *T. urticae* adults collected from bean and cucumber plants grown in commercial greenhouses (Pakdasht County in Tehran Province). The mites were maintained on detached bean leaves, placed with their abaxial side up on a moist sponge in a plastic container (dimensions: 15 × 10 × 5 cm). The sponge was moistened with tap water (approximately 80 mL). Strips of damp tissue paper surrounded the edges of the sponge to provide higher humidity and prevent the mites from escaping. A hole (diameter: 2 cm) was drilled into the container lid and covered with fine mesh cloth (pore size 0.053 μm) to ensure ventilation of cage while preventing mites from escaping. The containers were kept in a climatized chamber under constant conditions (temperature 25 ± 0.5 °C, 75 ± 5% RH, and a 16L:8D photoperiod). The leaf populated with *P. persimilis* was transferred to a new container every second day, and the colony of these mites that prey on *T. urticae* was maintained for at least three months before being used in bioassays.

### 4.2. Experimental Arena

*Phaseolus vulgaris* L. cv. Sunray seedlings were grown in plastic pots (15 cm in diameter) in a climatized chamber (temperature 25 ± 0.5 °C, 75 ± 5% RH, and a 16L:8D photoperiod) until they reached their fourth or fifth leaf stage. A leaf disk (3 cm in diameter) cut from bean leaves was placed abaxial side up on a layer of sponge moistened with 10 mL of tap water in a Petri dish (6 cm in diameter, 1 cm in height). The edges of the sponge were covered with moist tissue paper, which served as a barrier to prevent mites from escaping and maintained high humidity. To ensure ventilation of the area, a hole (2 cm in diameter) was cut in the center of the lid and covered with a fine mesh (pore size 0.053 μm), which prevented the mites from escaping.

### 4.3. Experimental Design

Laboratory experiments were conducted under three constant photoperiods, 8:16 h, 12:12 h, and 16:8 h (L:D), in environmental chambers (Binder KBWS 240, Tuttlingen, Germany) with 75 ± 5% relative humidity and a temperature of 25 ± 0.5 °C. To assess how the photoperiod affects the predation rate, *P. persimilis* eggs (<24 h old) were individually transferred into the experimental arenas. Eggs were laid by twenty pairs of 2-day-old female predators maintained under the conditions specified above. For each photoperiod treatment, the experiments started with 60 *P. persimilis* eggs. Hatched predatory mites were reared individually in the test arenas and fed 30 *T. urticae* eggs, collected from the stock culture with a fine camel hair brush, each day. The predation rate and survival of *P. persimilis* were checked daily, and any uneaten *T. urticae* eggs were removed and replaced with 30 new eggs, together with a fresh bean leaf. Upon reaching adulthood, males and females were paired for three days to allow mating. Each pair of *P. persimilis* was supplied with 75 *T. urticae* eggs daily. The predation rate and survival of the adults were recorded every 24 h until the death of the last predatory mite.

### 4.4. Statistical Analysis

The predation rate of *P. persimilis* under different photoperiods was estimated following the methodology outlined by Chi and Yang [44]. CONSUM-MSChart software (Version 26 Jun 2024) [53] and the age–stage, two-sex life table theory [54] were used for the analysis. Standard errors and variances of the calculated parameters were estimated using the bootstrap resampling method, with 100,000 iterations conducted [55]. Statistical differences between experimental groups were assessed using the paired bootstrap test. Differences with values of *p* < 0.05 were considered to be statistically significant. Table 4 presents the computed predation parameters and their respective equations and definitions.

## Figures and Tables

**Figure 1 plants-14-00687-f001:**
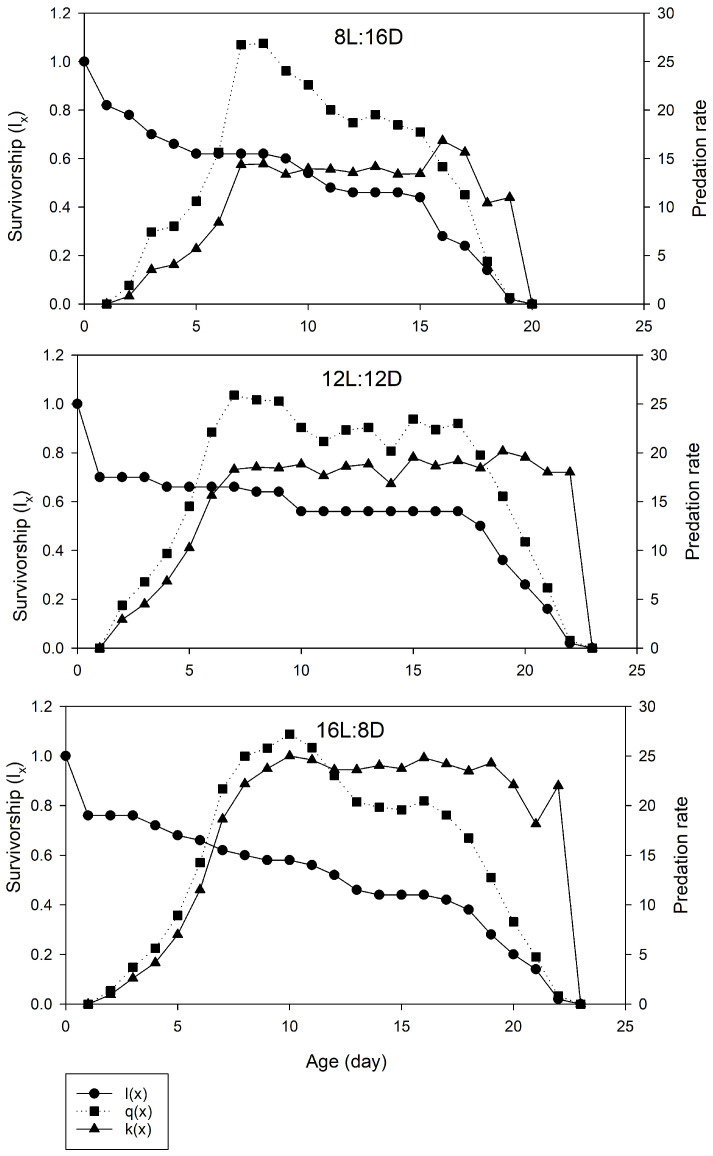
Age-specific survival rate (*l_x_*), age-specific predation rate (*k_x_*), and age-specific net predation rate (*q_x_*) of *Phytoseiulus persimilis* reared on eggs of *Tetranychus urticae* under different L:D photoperiods.

**Figure 2 plants-14-00687-f002:**
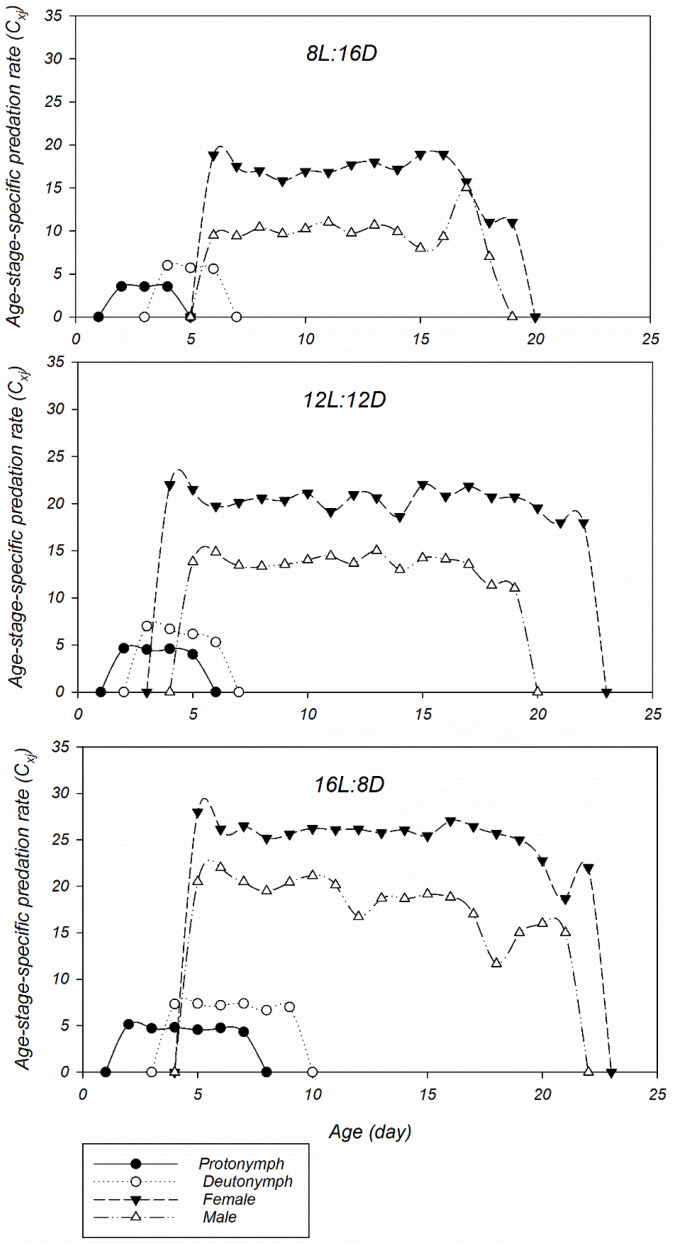
Age-stage-specific predation rate (*c_xj_*) of *Phytoseiulus persimilis* reared on eggs of *Tetranychus urticae* under different photoperiods.

**Table 1 plants-14-00687-t001:** Daily predation rate (*D_j_*) of *Phytoseiulus persimilis* reared on *Tetranychus urticae* eggs under different photoperiods.

Developmental Stage	8L:16D		12L:12D		16L:8D	
	*D_j_* ± S.E. *	n	*D_j_* ± S.E. *	n	*D_j_* ± S.E. *	n
Protonymph	3.54 ± 0.21 b	35	4.55 ± 0.23 a	35	4.74 ± 0.18 a	38
Deutonymph	5.70 ± 0.34 c	31	6.27 ± 0.31 b	33	7.26 ± 0.28 a	30
Female (adult)	17.02 ± 1.00 c	19	20.43 ± 1.00 b	24	25.66 ± 1.00 a	23
Male (adult)	9.91 ± 0.58 c	12	13.76 ± 0.67 b	9	18.85 ± 0.74 a	7

* Values followed by different letters within the same row are significantly different (paired bootstrap test, *p* < 0.05).

**Table 2 plants-14-00687-t002:** Predation rate of the different life stages (*P_j_*) of *Phytoseiulus persimilis* reared on *Tetranychus urticae* eggs under different photoperiods.

Developmental Stage	8L:16D		12L:12D		16L:8D	
	*P_j_* ± S.E. *	n	*P_j_* ± S.E. *	n	*P_j_* ± S.E. *	n
Protonymph	7.06 ± 0.17 c	31	8.7 ± 0.42 b	33	9.7 ± 0.36 a	30
Deutonymph	11.23 ± 0.32 a	31	9.12 ± 0.53 b	33	11.37 ± 0.63 a	30
Preadult	15.65 ± 0.60 c	31	18.83 ± 0.78 b	33	22 ± 0.70 a	30
Female (adult)	145.16 ± 4.36 b	19	261.29 ± 4.90 a	24	282.22 ± 4.80 a	23
Male (adult)	94.17 ± 3.46 c	21	181.89 ± 3.00 a	9	210 ± 2.65 a	7

* Values followed by different letters within the same row are significantly different (paired bootstrap test, *p* < 0.05).

**Table 3 plants-14-00687-t003:** Predation parameters of *Phytoseiulus persimilis* adult females reared on *Tetranychus urticae* eggs under different photoperiods.

Parameter	8L:16D *	12L:12D *	16L:8D *
Net predation rate, *C*_0_ (prey/predator)	89.77 ± 12.38 b	170.28 ± 20.47 a	173.22 ± 22.61 a
Finite predation rate, ω (d^−1^)	5.02 ± 0.42 c	6.55 ± 0.45 b	7.08 ± 0.59 a
Transformation rate, *Q_p_*	11.04 ± 1.48 b	7.99 ± 0.79 c	14.37 ± 1.50 a
Stable predation rate (ψ) (prey/predator)	4.21 ± 0.29 c	5.09 ± 0.28 b	5.84 ± 0.42 a

* Values followed by different letters within the same row are significantly different (paired bootstrap test, *p* < 0.05).

**Table 4 plants-14-00687-t004:** Population parameters and their respective definitions and equations used in the estimation of predation rates.

**Parameter**	Equation	Definition
Age-specific predation rate	$k_{x}=\frac{\sum_{j=1}^{\beta }s_{xj}c_{xj}}{\sum_{j=1}^{\beta }s_{xj}}$	The number of prey consumed by the surviving predators at age *x*.
Age–stage-specific predation rate	$c_{xj}=\frac{\sum_{i=1}^{n_{xj}}d_{xj,i}}{n_{xj}}$	The number of prey consumed by predators at age *x* and stage *j*.
Age-specific net predation rate	$q_{x}=l_{x}k_{x}=\sum_{j=1}^{\beta }s_{xj}c_{xj}$	The mean number of prey consumed by an individual at age *x*.
Net predation rate	$C_{0}=\sum_{x=0}^{\delta }\sum_{j=1}^{\beta }s_{xj}c_{xj}$	The total number of prey consumed by an individual during its life span.
Transformation rate	$Q_{p}=\frac{C_{0}}{R_{0}}$	The number of prey consumed to produce one offspring.
Predation rate for the stage	$P_{j}=\frac{\sum_{i=1}^{n_{j}}P_{ij}}{n_{j}}$	The predation rate of each predator in stage *j*.
Daily predation rate	$D_{j}=\frac{\sum_{x=a}^{b}c_{xj}s_{xj}}{\sum_{x=a}^{b}s_{xj}}$	The daily predation rate per predator in stage *j*.
Finite predation rate	$\omega =\lambda \sum_{i=0}^{\infty }\sum_{j=1}^{\beta }a_{xj}c_{xj}$	*a_xj_* indicates the proportion of individuals at age *x* and stage *j*.
Stable predation rate	$\varphi =\sum_{i=0}^{\infty }\sum_{j=1}^{\beta }a_{xj}c_{xj}$	The total predation capacity of a stable population.

## Data Availability

The data that support the findings of this study are openly available on FIGSHARE at https://doi.org/10.6084/m9.figshare.28030478.v1 (accessed on 16 December 2024).

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
