# Peer review of "The Effect of Light Cycles on the Predation Characteristics of Phytoseiulus persimilis (Acari: Phytoseiidae) Feeding on Tetranychus urticae (Acari: Tetranychidae)"

_plants, 2025, doi:10.3390/plants14050687_

Round 1

Reviewer 1 Report

Comments and Suggestions for Authors

The paper is well written and the experiments are well planned and done.

Just few minor comments:

  • line 109-110: the sentence seems to be a conclusion and it seems to be out of order;
  • line 211: a ")" was skipped after photoperiod;
  • line 217: thin vein is vague;
  • line 221: increased should be wrong

Author Response

Comment 1: The paper is well written and the experiments are well planned and done.

Response 1: Thank you for a positive review report.

Just few minor comments:

Comment 2: line 109-110: the sentence seems to be a conclusion and it seems to be out of order;

Response 2: This sentence was deleted and text on line 108 was modified.

Comment 3: line 211: a ")" was skipped after photoperiod;

Response 3: Corrected.

Comment 4: line 217: thin vein is vague;

Response 4: Information about vein thickness as well as previous sentence were deleted because of redundancy.

Comment 5: line 221: increased should be wrong

Response 5: Corrected.

Reviewer 2 Report

Comments and Suggestions for Authors

Study aims to examine the influence of photoperiod on predation rate of a predatory mite (Phytoseiulus persimilis) as this mite commonly occurs in crops worldwideand is known to contribute to control of pest mites especially Tetranychus urticae. Its success has led to commercial rearing in insectaries for release in green houses and field crops. Results identify the photoperiod associated with higher levels of predation. Identifying factors to maximize effectiveness provides valuable information. Results are of particular relevance in green houses where control of photoperiod is possible-less clear regarding role of such information in field settings. The authors mention field crops and could explain relevance of study outcome to field crops. (abstract L 24 16L:8D photoperiod can thus be recommended as optimal for biocontrol of spider mites in both field and greenhouse settings). The manuscript is well written, methods clearly explained and results robust and presented with helpful graphs and tables. While the experiments and outcome are robust and informative , more discussion of the intersection of this system with agricultural rather than laboratory or insectary scenarios should be included.

The authors report predation highest with longer period of light which indicates optimal light period. Please provide some information regarding greenhouse photoperiods-where the experimental times chosen in the experiments related to common greenhouse photoperiod settings? And indicate how the identification of longer light cycles could be useful in field crops including orchards.

Some discussion of potential causes for increased predation with longer light periods would be useful-some points of interest are raised in Kazak et al (2004). The references to ladybird feeding invoke visual cues underlying increased predation in ladybirds as a response to longer periods of light (Wang et al  2013; Tan et al 2014) but this is probably not the case for predatory mites.

Ll36-39 please explain difference between ‘outdoor and indoor environments ‘and ‘orchard and greenhouse crops’? or edit to clarify. Reference (6) does not support the claim made in the sentence preceding it. There are better references  to the claim of success of these mites in biocontrol (7-10) with the exception of (10).

L50 please provide more relevant reference for statement re length of daylight impacting predation rate only one refers to mites (20) (others are ladybirds) and this study refers to host plant impacts and was conducted at 16:8 (L:D).

L193 ‘16L:8D photoperiod identified as optimal for the release of this biocontrol agent in the management of two-spotted spider mites in both field and greenhouse conditions’. Please explain relevance to field conditions

For clarity please either the year for the publication or referral to reference number when results of previously published work are mentioned eg L140 Wang et al (2014) or (16); L142 Tan et al (2014) or (25)

L 141, 143 and elsewhere Coccinellidae not Coccinillidae

Author Response

Comment 1: Study aims to examine the influence of photoperiod on predation rate of a predatory mite (Phytoseiulus persimilis) as this mite commonly occurs in crops worldwideand is known to contribute to control of pest mites especially Tetranychus urticae. Its success has led to commercial rearing in insectaries for release in green houses and field crops. Results identify the photoperiod associated with higher levels of predation. Identifying factors to maximize effectiveness provides valuable information. Results are of particular relevance in green houses where control of photoperiod is possible-less clear regarding role of such information in field settings. The authors mention field crops and could explain relevance of study outcome to field crops. (abstract L 24 16L:8D photoperiod can thus be recommended as optimal for biocontrol of spider mites in both field and greenhouse settings). The manuscript is well written, methods clearly explained and results robust and presented with helpful graphs and tables. While the experiments and outcome are robust and informative , more discussion of the intersection of this system with agricultural rather than laboratory or insectary scenarios should be included.

Response 1: We appreciate positive review of our manuscript and the problem of recommendation of optimal photoperiod for field crops. We therefore modified this statement and limited it to greenhouse conditions. The end of discussion was more elaborated.

Comment 2: The authors report predation highest with longer period of light which indicates optimal light period. Please provide some information regarding greenhouse photoperiods-where the experimental times chosen in the experiments related to common greenhouse photoperiod settings? And indicate how the identification of longer light cycles could be useful in field crops including orchards.

Response 2: Some information on this topic was added into Introduction at the end of 3rd paragraph. Since it is unlikely to manage photoperiod in field crops we modified the text of the manuscript to include only controlled environments/greenhouses.

Comment 3: Some discussion of potential causes for increased predation with longer light periods would be useful-some points of interest are raised in Kazak et al (2004). The references to ladybird feeding invoke visual cues underlying increased predation in ladybirds as a response to longer periods of light (Wang et al  2013; Tan et al 2014) but this is probably not the case for predatory mites.

Response 3: Thank you for this comment. Some text was added at the end of 2nd paragraph of Discussion with references added and potential causes explaining higher predation a longer photoperiod.

Comment 4: Ll36-39 please explain difference between ‘outdoor and indoor environments ‘and ‘orchard and greenhouse crops’? or edit to clarify. Reference (6) does not support the claim made in the sentence preceding it. There are better references  to the claim of success of these mites in biocontrol (7-10) with the exception of (10).

Response 4: This part of text was rewritten and relevant references were added while those which do not suppor the claims were removed.

Comment 5: L50 please provide more relevant reference for statement re length of daylight impacting predation rate only one refers to mites (20) (others are ladybirds) and this study refers to host plant impacts and was conducted at 16:8 (L:D).

Response 5: Reference [20] was replaced with more relevant to the effect of photoperiod on predation of predatory mites.

Comment 6: L193 ‘16L:8D photoperiod identified as optimal for the release of this biocontrol agent in the management of two-spotted spider mites in both field and greenhouse conditions’. Please explain relevance to field conditions

Response 6: Field conditions were removed because it is unlikely anybody could control light conditions there.

Comment 7: For clarity please either the year for the publication or referral to reference number when results of previously published work are mentioned eg L140 Wang et al (2014) or (16); L142 Tan et al (2014) or (25)

Response 7: Numerical references were placed after author names in these and other places where authors are named.

Comment 8: L 141, 143 and elsewhere Coccinellidae not Coccinillidae

Response 8: We appologize for these typos, it is now corrected.

Reviewer 3 Report

Comments and Suggestions for Authors

I made the review of the manuscript plants-3484967 and I found the work very interesting and well written.  I made just few editorial suggestions, related to the fact I do not think it is necessary to put the Author's names  (and the Order and Family) in the abstract, while it is mandatory in the Introduction chapter. 

The only important comment is at the end of the discussion (the only point where the Authors mentioned the possible side effects of the changes in photoperiodism on  phytophagous arthropods: my suggestion is to give more space to this aspect, very important (in our laboratories we are controlling early stage T. urticae infestations  playing with the amount of light (see my comments in the PDF). 

Author Response

Comment 1: I made the review of the manuscript plants-3484967 and I found the work very interesting and well written.  I made just few editorial suggestions, related to the fact I do not think it is necessary to put the Author's names  (and the Order and Family) in the abstract, while it is mandatory in the Introduction chapter. 

Response 1: Thank you for editorial suggestions; we agree and removed these details from Abstract and placed them in Introduction.

Comment 2: The only important comment is at the end of the discussion (the only point where the Authors mentioned the possible side effects of the changes in photoperiodism on  phytophagous arthropods: my suggestion is to give more space to this aspect, very important (in our laboratories we are controlling early stage T. urticae infestations  playing with the amount of light (see my comments in the PDF). 

Response 2: The effect of long-day photoperiod on T. urticae is now discussed and implications for IPM recommendations included.

Reviewer 4 Report

Comments and Suggestions for Authors

A brief summary

Phytoseiulus persimilis is an important natural enemy for spider mites. The use of this species is encouraged for biological control of spider mites in the scheme of integrated pest management. The authors of this manuscript indicated its importance for managing Tetranychus urticae by studying the effect of three different photoperiods on predation of P. persimillis on T.urticae. The manuscript is fairly short, pretty straightforward, easy to understand and easy to follow. The figures helped to convey and clarify the information in the results. The authors explained well on the ultimate goal of conducting the study, i.e. to improve the application timing in greenhouse or field. However, there are some missing details that should be included. Definitely, there is room for improvement to strengthen the content and a few suggested changes are listed below.

Specific comments

Title (p. 1)

Lines 1-3

The two families put together at the end of the title is confusing. It should be split up right next to the species name. See below.

Phytoseiulus persimilis (Acari: Phytoseiidae) Feeding … Tetranychus urticae (Acari: Tetranychidae)

Introduction (p. 2)

Lines 69-70

Your sentence indicated that you are for sure now there was no information, but maybe someone has done the study before but just not published, or you did not have access to such information. Therefore, I suggest the sentence is rephrased as below or something like this:

    However, to our knowledge the information on the … is currently unavailable.

Results (p. 4)

Line 124

Figure 1 à the third figure missing L and D; should have been 16L: 8D.

Materials and Methods (p. 7-8)

Line 204

… in greenhouses (…) à please be more specific whether it is commercial greenhouse, university greenhouse etc.

Line 206

How do you moist the sponge? Do you moist with water? If so, what kind of water and how much (volume)?

Line 211

Need “)” after the word ‘photoperiod’.

Line 216

What is the temperature, relative humidity, and photoperiod in the climatized chamber?

Line 217

Tetranychus urticae should be abbreviated as T. urticae since it is already spelled out previously.

Line 220

How do you moist the tissue paper? Do you moist with water? If so, what kind of water and how much (volume)?

Line 222

What size of fine mesh? Please be more specific.

Lines 225-230

It was not clear if the experiments were replicated. It needs more clarification.

Line 231

… separate incubators … à list the manufacturer next to the word “incubators” in parentheses.

Lines 233-236

The sentence is a bit too long and confusing. Please reword.

Line 240

Replace the word “data” with “statistical” à Statistical Analysis (instead of Data Analysis)

Line 245

Add information on what is considered as statistically significant.

Author Response

A brief summary

Comment 1: Phytoseiulus persimilis is an important natural enemy for spider mites. The use of this species is encouraged for biological control of spider mites in the scheme of integrated pest management. The authors of this manuscript indicated its importance for managing Tetranychus urticae by studying the effect of three different photoperiods on predation of P. persimillis on T.urticae. The manuscript is fairly short, pretty straightforward, easy to understand and easy to follow. The figures helped to convey and clarify the information in the results. The authors explained well on the ultimate goal of conducting the study, i.e. to improve the application timing in greenhouse or field. However, there are some missing details that should be included. Definitely, there is room for improvement to strengthen the content and a few suggested changes are listed below.

Response 1: Thank you for positive review and suggestions for improvement which we incorporated into the revised manuscript.

Specific comments

Title (p. 1)

Lines 1-3

Comment 2: The two families put together at the end of the title is confusing. It should be split up right next to the species name. See below.

Phytoseiulus persimilis (Acari: Phytoseiidae) Feeding … Tetranychus urticae (Acari: Tetranychidae)

Response 2: Thank you for this suggestion, we fully agree and made changes accordinly.

Introduction (p. 2)

Lines 69-70

Comment 3: Your sentence indicated that you are for sure now there was no information, but maybe someone has done the study before but just not published, or you did not have access to such information. Therefore, I suggest the sentence is rephrased as below or something like this:

    However, to our knowledge the information on the … is currently unavailable.

Response 3: Text was rewritten as suggested.

Results (p. 4)

Line 124

Comment 4: Figure 1 à the third figure missing L and D; should have been 16L: 8D.

Response 4: Thank you for noticing this mistake which we corrected in revised version.

Materials and Methods (p. 7-8)

Line 204

Comment 5: … in greenhouses (…) à please be more specific whether it is commercial greenhouse, university greenhouse etc.

Response 5: They were commercial greenhouses. This detail was added into text.

Line 206

Comment 6: How do you moist the sponge? Do you moist with water? If so, what kind of water and how much (volume)?

Response 6: More details were included in the description of methods.

Line 211

Comment 7: Need “)” after the word ‘photoperiod’.

Response 7: Corrected.

Line 216

Comment 8: What is the temperature, relative humidity, and photoperiod in the climatized chamber?

Response 8: Such information was added.

Line 217

Comment 9: Tetranychus urticae should be abbreviated as T. urticae since it is already spelled out previously.

Response 9: This was modified to abbreviated form of species name.

Line 220

Comment 10: How do you moist the tissue paper? Do you moist with water? If so, what kind of water and how much (volume)?

Response 10: These details were added into the text.

Line 222

Comment 11: What size of fine mesh? Please be more specific.

Response 11: This detail was added into the text.

Lines 225-230

Comment 12: It was not clear if the experiments were replicated. It needs more clarification.

Response 12: For each photoperiod treatment, the experiments started with 60 P. persimilis eggs. The text was therefore modified for clarity. Please note, that n values in tables are lower because in each stage mites suffered from mortality which gradually decreased the mite numbers.

Line 231

Comment 13: … separate incubators … à list the manufacturer next to the word “incubators” in parentheses.

Response 13: There were grow chambers already mentioned above in the text but we revised this part for better clarity.

Lines 233-236

Comment 14: The sentence is a bit too long and confusing. Please reword.

Response 14: This paragraph was rewritten for better clarity.

Line 240

Comment 15: Replace the word “data” with “statistical” à Statistical Analysis (instead of Data Analysis)

Response 15: Text was replaced as suggested. 

Line 245

Comment 16: Add information on what is considered as statistically significant.

Response 16: The sentence “Differences with values of P<0.05 were considered to be statistically significant.” was added.